# DIFFUSION MODELS IN SPACE AND TIME VIA THE DISCRETIZED HEAT EQUATION

## ABSTRACT

We propose a new class of diffusion models which use noising processes that diffuse jointly in space and time. These noising processes evolve according to a stochastic differential equation (SDE) inspired by the heat equation, a canonical space-time diffusion. We show that sampling from the diffusion's transition density and evaluating its score remain tractable in the Fourier domain. This approach smooths the sequence of distributions that bridge noise and data, decaying high-frequency information before the lower frequencies that encode the large-scale structure of the image. We evaluate these models on MNIST and find that they generate convincing samples.

## 1 INTRODUCTION

Diffusion models are a promising class of generative models which achieve high quality samples and likelihoods across a variety of domains (Dhariwal & Nichol, 2021; Chen et al., 2020; Luo & Hu, 2021). In these models, a fixed noising process gradually converts data to unstructured noise, then a learned denoising process converts this noise back into data (Sohl-Dickstein et al., 2015; Ho et al., 2020; Song et al., 2020). One method to learn the denoising process estimates the score (the gradient of the log probability with respect to the data) of each of the noise-perturbed distributions along the noising path. This score is then used to define a stochastic differential equation (SDE) which reverses the noising process when run backwards in time.

The more smoothly the noising process interpolates between the data distribution and the noise distribution, the easier it is to accurately estimate the score, yielding benefits in training time, synthesis quality, and sampling speed. Commonly used noising processes are smooth in time but ignore any spatial structure in the data, e.g. nearby pixel correlations in an image.

We propose to incorporate the spatial structure of data by diffusing jointly in space and time. The canonical space-time diffusion is given by the heat equation, a partial differential equation describing the evolution of heat in a medium. Inspired by this equation, we propose adding a discretized Laplacian term to the drift of the noising SDE. This term encourages the dispersal of "heat" in the image, resulting in a gradual blurring. High frequency components decay before low frequency ones, meaning that low-level detail is lost before the overall structure of the image. As the Laplacian is a linear operator, the resulting SDE remains linear and its transition densities at any given times are Gaussian. We show that it is possible to efficiently sample from the transition density and evaluate its score using the Fast Fourier Transform (FFT) (Cooley & Tukey, 1965), since the transition matrix and covariance of the transition density are diagonal in the Fourier domain.

We exhibit promising results on the MNIST dataset (LeCun et al., 2010) which shows our model can accurately capture the true data distribution.

## 2 BACKGROUND

### 2.1 DIFFUSION MODELS

Both the noising and the denoising process of a diffusion model can be described in continuous time as solutions to SDEs. Given data $u_0 \in \mathbb{R}^D$ drawn from the data distribution $\mathbb{P}_{\text{data}}$, we use the

following Itô SDE to define the forward noising process:

$$d\boldsymbol{u}_t = \boldsymbol{f}(\boldsymbol{u}_t, t)dt + g(t)d\boldsymbol{w}_t, \quad t \in [0, 1]$$

$\boldsymbol{f} : \mathbb{R}^D \times \mathbb{R}_+ \to \mathbb{R}^D$ is the drift coefficient of the SDE, $g : \mathbb{R}_+ \to \mathbb{R}$ is the diffusion coefficient, and $\{\boldsymbol{w}_t\}$ is a standard Wiener process. We can choose $\boldsymbol{f}$ and $g$ so that $\boldsymbol{u}_1 \mid \boldsymbol{u}_0 \sim \mathbb{P}_{\text{noise}}$ for some known Gaussian distribution $\mathbb{P}_{\text{noise}}$ regardless of $\boldsymbol{u}_0$ and therefore marginally $\boldsymbol{u}_1 \sim \mathbb{P}_{\text{noise}}$. Anderson (1982) showed that the reverse stochastic process $\bar{\boldsymbol{u}}_t = \boldsymbol{u}_{1-t}$ solves the SDE:

$$d\bar{\boldsymbol{u}}_t = \left[ \boldsymbol{f}(\bar{\boldsymbol{u}}_t, 1 - t) - g(1 - t)^2 \nabla_{\bar{\boldsymbol{u}}_t} \log p_{1-t}(\bar{\boldsymbol{u}}_t) \right] dt + g(1 - t)d\bar{\boldsymbol{w}}_t$$

where $p_t$ is the marginal density of $\boldsymbol{u}_t$. This allows us to sample from the data distribution as long as the score $\nabla_{\boldsymbol{u}} \log p_t(\boldsymbol{u})$ is known for all $t \in [0, 1]$: we start by drawing a sample $\bar{\boldsymbol{u}}_0 \sim \mathbb{P}_{\text{noise}}$, then approximately solve the reverse SDE so that $\bar{\boldsymbol{u}}_1 \sim \mathbb{P}_{\text{data}}$. We can estimate the score $\nabla_{\boldsymbol{u}} \log p_t(\boldsymbol{u})$ using a neural network $\boldsymbol{s}_{\boldsymbol{\theta}}(\boldsymbol{u}, t)$, where the network parameters $\boldsymbol{\theta}$ are found by minimizing the denoising score matching objective:

$$J(\boldsymbol{\theta}) = \mathbb{E}_{t \sim \text{Uni}(0,1)} \mathbb{E}_{\boldsymbol{u}_0 \sim \mathbb{P}_{\text{data}}} \mathbb{E}_{\boldsymbol{u}_t \sim p_{0t}(\boldsymbol{u}_t | \boldsymbol{u}_0)} \left[ \lambda(t) \| \boldsymbol{s}_{\boldsymbol{\theta}}(\boldsymbol{u}_t, t) - \nabla_{\boldsymbol{u}_t} \log p_{0t}(\boldsymbol{u}_t \mid \boldsymbol{u}_0) \|_2^2 \right] \quad (1)$$

Typically, we choose $\boldsymbol{f}(\boldsymbol{u}, t) = -f(t)\boldsymbol{u}$ where $f : \mathbb{R}^D \to \mathbb{R}_+$ is a non-nonnegative scalar valued function, leading to the SDE:

$$d\boldsymbol{u}_t = -f(t)\boldsymbol{u}_t dt + g(t)d\boldsymbol{w}_t, \quad t \in [0, 1] \quad (2)$$

## 2.2 The Heat Equation

The heat equation is a fundamental partial differential equation describing the evolution of heat in an idealized medium. Define $u_t : \mathbb{R}^2 \to \mathbb{R}$ so that $u_t(x, y)$ is the temperature of the point $(x, y)$ in the plane at time $t$. We say $u$ is a solution is a solution to the heat equation if:

$$\frac{\partial u}{\partial t} = \gamma \left( \frac{\partial^2 u}{\partial x^2} + \frac{\partial^2 u}{\partial y^2} \right)$$

$\gamma$ is the thermal diffusivity of the medium, with larger values of $\gamma$ leading to faster dispersion of heat. Defining the Laplacian operator $\Delta = \frac{\partial^2}{\partial x^2} + \frac{\partial^2}{\partial y^2}$, we can write the heat equation as:

$$\frac{\partial u}{\partial t} = \gamma \Delta u \quad (3)$$

The heat equation (3) describes the evolution of temperature in continuous space and time. However, an image is only defined on a discrete grid in space. In order to adapt the heat equation to this setting, we must discretize the Laplacian $\Delta$. It is standard to do so through the finite difference method:

$$(\Delta u)(x, y) \approx \frac{u(x + \epsilon, y) + u(x - \epsilon, y) + u(x, y + \epsilon) + u(x, y - \epsilon) - 4u(x, y)}{\epsilon^2}$$

We can easily adapt the finite difference operator to an image $\boldsymbol{u} \in \mathbb{R}^{D \times D}$ defined on a discrete grid, treating $\epsilon$ as the distance between adjacent points on the grid. We will choose $\epsilon = 1$ for convenience, resulting in the following discretized Laplacian operator[1] $\tilde{\Delta} : \mathbb{R}^{D \times D} \to \mathbb{R}^{D \times D}$:

$$(\tilde{\Delta}\boldsymbol{u})[x, y] = \boldsymbol{u}[x + 1, y] + \boldsymbol{u}[x - 1, y] + \boldsymbol{u}[x, y + 1] + \boldsymbol{u}[x, y - 1] - 4\boldsymbol{u}[x, y]$$

## 3 Diffusion Models in Space and Time

We propose to modify the standard noising diffusion defined by Equation 2 by adding a discretized Laplacian term to its drift, scaled by a time dependent thermal diffusivity $\gamma : \mathbb{R}_+ \to \mathbb{R}_+$:

$$d\boldsymbol{u}_t = \left[ -f(t)\boldsymbol{u}_t + \gamma(t)(\tilde{\Delta}\boldsymbol{u}_t) \right] dt + g(t)d\boldsymbol{w}_t \quad (4)$$

This additional term will lead to a smoother noising diffusion. Recall that $(\tilde{\Delta}\boldsymbol{u})[i, j]$ is the summed difference between pixel $\boldsymbol{u}[i, j]$ and each of its neighbors. Thus, including this term compels each

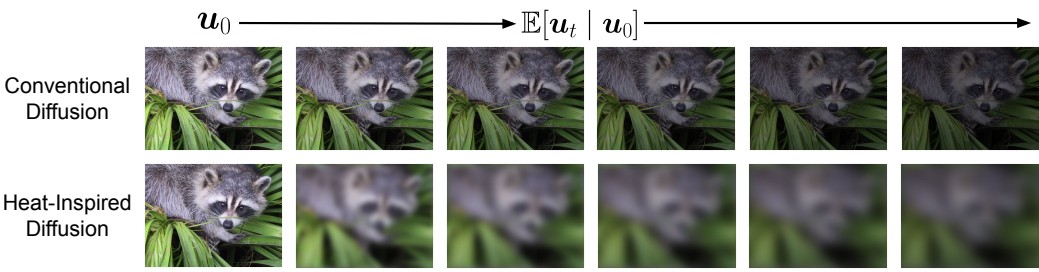

**Figure 1:** Visualization of the mean of the diffusion process over time, starting with image $\boldsymbol{u}_0$. Conventional diffusions simply contract the image uniformly towards the origin, while including the Laplacian term causes the image to blur and pixel intensities to spread from their original locations.

pixel to take on its neighbors values, resulting in a smooth diffusion in space (i.e. pixels) and time. We contrast the two approaches with and without the additional Laplacian term in Figure 1 above.

In order to use the diffusion process defined by Equation 4 in the score-based generative modeling framework, we must be able to efficiently sample from its transition density and evaluate this density's score at any given time. As the SDE is linear, we know the transition density is of the form $p_{st}(\boldsymbol{u}_t \mid \boldsymbol{u}_s) = \mathcal{N}(\boldsymbol{u}_t; \boldsymbol{A}(t,s)\boldsymbol{u}(s), \boldsymbol{P}(t,s))$ for some $\boldsymbol{A}, \boldsymbol{P}$ (Särkkä & Solin, 2019). In order to efficiently compute $\boldsymbol{A}$ and $\boldsymbol{P}$, we use that the linear operator $\tilde{\Delta}$ is diagonalized by the discrete Fourier transform. Indeed, given $k, l \in \mathbb{Z}^2$, let $\boldsymbol{F}_{k,l} \in \mathbb{C}^{D \times D}$ denote the associated Fourier mode:

$$\boldsymbol{F}_{k,l}[x,y] = \exp\left(\frac{2\pi i}{D}xk + \frac{2\pi i}{D}yl\right)$$

Then $\boldsymbol{F}_{k,l}$ is an eigenvector of $\tilde{\Delta}$ with eigenvalue $\lambda_{k,l}$ (see Appendix A):

$$\tilde{\Delta}\boldsymbol{F}_{k,l} = \underbrace{\left(2\cos\left(\frac{2\pi}{D}k\right) + 2\cos\left(\frac{2\pi}{D}l\right) - 4\right)}_{\lambda_{k,l}} \boldsymbol{F}_{k,l}$$

Thus, the Fourier modes $\{\boldsymbol{F}_{k,l} : k, l \in \{-\frac{D}{2}+1, ..., 0, ..., \frac{D}{2}\}^2\}$ comprise an orthogonal basis of eigenvectors for $\tilde{\Delta}$, where we have assumed $D$ is even. This observation allows us to tractably evaluate $\boldsymbol{A}$ and $\boldsymbol{P}$, as both are diagonal in frequency space.

**Theorem 1.** *Let $\mathcal{F}: \mathbb{R}^{D \times D} \to \mathbb{C}^{D \times D}$ denote the discrete two-dimensional Fourier transform, and let $\mathcal{F}^{-1}$ denote its inverse. For simplicity, we can view $\mathcal{F}$ as a linear map from $\mathbb{R}^{D^2} \to \mathbb{C}^{D^2}$ by stacking the columns of a matrix into a single vector and likewise for $\mathcal{F}^{-1}$. This allows us to abuse notation and interpret $\mathcal{F}$ as a $\mathbb{C}^{D^2} \times \mathbb{C}^{D^2}$ matrix.*

*Let $\mathrm{vec}: \mathbb{R}^{D \times D} \to \mathbb{R}^{D^2}$ convert a matrix into a vector by stacking its columns. We claim the SDE given by Equation 4 has transition density:*

$$p_{st}(\boldsymbol{u}_t \mid \boldsymbol{u}_s) = \mathcal{N}(\boldsymbol{u}_t; \boldsymbol{A}(t,s)\boldsymbol{u}_s, \boldsymbol{P}(t,s))$$

*where:*

$$\boldsymbol{A}(t,s) = \mathcal{F}^{-1}\mathrm{diag}\left(\mathrm{vec}\left(\underbrace{\left[\exp\left(\int_s^t \lambda_{k,l}\gamma(\tau) - f(\tau)d\tau\right)\right]_{k,l}}_{\boldsymbol{\Psi}(t,s)}\right)\right)\mathcal{F} \quad (5)$$

$$\boldsymbol{P}(t,s) = \mathcal{F}^{-1}\mathrm{diag}\left(\mathrm{vec}\left(\underbrace{\left[\int_s^t g(\tau)^2 \exp\left(2\int_\tau^t \lambda_{k,l}\gamma(r) - f(r)dr\right)d\tau\right]_{k,l}}_{\boldsymbol{\Sigma}(t,s)}\right)\right)\mathcal{F} \quad (6)$$

*Proof.* See Appendix B. $\square$

---

[1] $\tilde{\Delta}$ is extended to the boundary edges by assuming $\boldsymbol{u}$ is periodic.

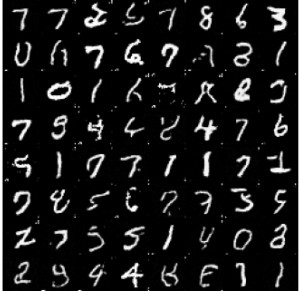

**Figure 2:** Samples from a trained model on the MNIST dataset.

Writing the transition density in this form gives additional insight into our heat-inspired SDE given in Equation 4. Note that $\lambda_{k,l} < 0$ for all $k, l$ such that $k \neq 0$ or $l \neq 0$, and that $\lambda_{k,l}$ is larger in absolute magnitude for $k, l$ which have larger absolute magnitude. Thus, in the Fourier domain, diffusing using our proposed SDE will attenuate the larger frequencies of an image more strongly than its smaller ones, a well-known consequence of the heat equation. Low level details of the image will be lost first, while high level details will be preserved later in the diffusion process.

The score-matching objective in Equation 1 requires sampling $\boldsymbol{u}_t \sim p_{0t}(\boldsymbol{u}_t \mid \boldsymbol{u}_0)$ and evaluating the score $\nabla_{\boldsymbol{u}_t} \log p_{0t}(\boldsymbol{u}_t \mid \boldsymbol{u}_0)$ of the sample. Using Theorem 1, we show how this can be done efficiently using the FFT and the inverse FFT in Algorithm 1.

---

**Algorithm 1** Efficient Sampling from the Heat-Inspired Transition Density

---

1: **Input:** Initial image $\boldsymbol{u}_0 \in \mathbb{R}^{D \times D}$, time $t \in \mathbb{R}_+$
2: Compute $\boldsymbol{\Psi}(t,0) = \left[\exp\left(\int_0^t \lambda_{k,l}\gamma(\tau) - f(\tau)d\tau\right)\right]_{k,l} \in \mathbb{R}^{D \times D}$ using numerical integration or known antiderivatives
3: Compute $\boldsymbol{\Sigma}(t,0) = \left[\int_0^t g(\tau)^2 \exp\left(2\int_\tau^t \lambda_{k,l}\gamma(r) - f(r)dr\right)d\tau\right]_{k,l} \in \mathbb{R}^{D \times D}$ using numerical integration or known antiderivatives
4: Compute mean $\hat{\boldsymbol{m}}_t$ in frequency space: $\hat{\boldsymbol{m}}_t = \boldsymbol{\Psi}(t,0) \odot \mathcal{F}(\boldsymbol{u}_0)$
5: Sample $\boldsymbol{\epsilon} \sim \mathcal{N}(\mathbf{0}, \boldsymbol{I})$, and compute the scaled frequency noise $\hat{\boldsymbol{\epsilon}} = \boldsymbol{\Sigma}(t,0)^{1/2} \odot \mathcal{F}(\boldsymbol{\epsilon})$
6: Compute the score in frequency space: $\hat{\boldsymbol{s}} = \boldsymbol{\Sigma}(t,0)^{-1/2} \odot \mathcal{F}(\boldsymbol{\epsilon})$
7: **Return:** Sample $\boldsymbol{u}_t = \mathcal{F}^{-1}(\hat{\boldsymbol{m}}_t + \hat{\boldsymbol{\epsilon}})$ and sample's score $\mathcal{F}^{-1}(\hat{\boldsymbol{s}})$

---

## 4 RELATED WORK

Our approach can be interpreted as learning a *stochastic* inverse to the heat equation. Running the heat equation backwards in time to sharpen an image is a well-studied problem in image analysis (Lindenbaum et al., 1994; Hummel et al., 1987; Vese & Le Guyader, 2016). Naive approaches to reversing the heat equation are numerically unstable, as small errors in the starting point become exponentially large when run backwards in time. The novelty in our approach is that we learn a *stochastic* inverse to the heat equation, thus avoiding this instability. Preliminary investigation showed that accurate estimation of the score is essential, otherwise the reverse SDE falls prey to the same instabilities of the deterministic reverse heat equation.

## 5 EXPERIMENTS

We trained our proposed diffusion model on the MNIST dataset of handwritten digits. In these preliminary experiments, our main goal was to see if the model could generate convincing samples. We leave for future work a careful comparison of our heat-inspired diffusion to conventional diffusion processes to assess whether training is easier or synthesis quality is improved.

Samples from the trained model are shown in Figure 2. We include architectural and training details in Appendix C. We see that the model is able to capture many features of the MNIST dataset. Most samples are recognizable as digits, and the model accurately places white symbols on a deep black background. However, there are some samples which are clearly not from the data distribution.

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

## A  THE DISCRETIZED LAPLACIAN IS DIAGONALIZED BY THE FOURIER TRANSFORM

In this section, we show that:

$$\tilde{\Delta}\boldsymbol{F}_{k,l} = \underbrace{\left( 2\cos\left(\frac{2\pi}{D}k\right) + 2\cos\left(\frac{2\pi}{D}l\right) - 4 \right)}_{\lambda_{k,l}} \boldsymbol{F}_{k,l}$$

Taking indices modulo $D$, we have:

$$\tilde{\Delta}\boldsymbol{F}_{k,l}[x,y] = \boldsymbol{F}_{k,l}[x+1,y] + \boldsymbol{F}_{k,l}[x-1,y] + \boldsymbol{F}_{k,l}[x,y+1] + \boldsymbol{F}_{k,l}[x,y-1] - 4\boldsymbol{F}_{k,l}[x,y]$$
$$= \exp\left(\frac{2\pi i}{D}xk + \frac{2\pi i}{D}yl\right)\left\{\exp\left(\frac{2\pi i}{D}k\right) + \exp\left(-\frac{2\pi i}{D}k\right) + \exp\left(\frac{2\pi i}{D}l\right) + \exp\left(-\frac{2\pi i}{D}l\right) - 4\right\}$$
$$= \boldsymbol{F}_{k,l}[x,y]\left\{2\cos\left(\frac{2\pi}{D}k\right) + 2\cos\left(\frac{2\pi}{D}l\right) - 4\right\}$$
$$= \lambda_{k,l}\boldsymbol{F}_{k,l}[x,y]$$

## B  THEOREM 1 (TRANSITION DENSITY PARAMETERS ARE DIAGONAL IN FREQUENCY SPACE)

We begin by finding the transition operator $\boldsymbol{A}(t,s)$ for the associated linear ordinary differential equation (ODE) without injected noise:

$$d\boldsymbol{u}_t = \left[-f(t)\boldsymbol{u}_t + \gamma(t)(\tilde{\Delta}\boldsymbol{u}_t)\right]dt \tag{7}$$

This is the operator such that $\boldsymbol{u}_t = \boldsymbol{A}(t,s)\boldsymbol{u}_s$ solves the above ODE with given initial condition $\boldsymbol{u}_s$ at time $s$. Throughout this proof, we will view $\boldsymbol{u}_t \in \mathbb{R}^{D^2}$ for simplicity, where we use vec to convert matrices to vectors by stacking their columns. Let $\boldsymbol{\Lambda} \in \mathbb{R}^{D^2 \times D^2}$ be the diagonal matrix which multiplies by the eigenvalues of $\tilde{\Delta}$:

$$\boldsymbol{\Lambda} = \operatorname{diag}\left(\operatorname{vec}\left([\lambda_{k,l}]_{k,l}\right)\right)$$

Since the Fourier transform diagonalizes $\tilde{\Delta}$, we have:

$$\tilde{\Delta} = \mathcal{F}^{-1}\boldsymbol{\Lambda}\mathcal{F}$$

We can therefore rewrite the differential equation above as:

$$d\boldsymbol{u}_t = \left[\mathcal{F}^{-1}(-f(t)\boldsymbol{I} + \gamma(t)\boldsymbol{\Lambda})\mathcal{F}\boldsymbol{u}_t\right]dt$$

We see that the ODE is diagonal in frequency space. If we let $\boldsymbol{w}_t = \mathcal{F}(\boldsymbol{u}_t)$, $\boldsymbol{w}_t$ satisfies the ODE:

$$d\boldsymbol{w}_t = \left[\mathcal{F}\mathcal{F}^{-1}(\gamma(t)\boldsymbol{\Lambda} - f(t)\boldsymbol{I})\mathcal{F}\boldsymbol{u}_t\right]dt$$
$$= (\gamma(t)\boldsymbol{\Lambda} - f(t)\boldsymbol{I})\boldsymbol{w}_t dt$$

As this decomposes into a decoupled system of scalar linear ODEs, we know the solution to this ODE with initial value $\boldsymbol{w}_s$ is:

$$\boldsymbol{w}_t = \operatorname{diag}\left(\operatorname{vec}\left(\left[\exp\left(\int_s^t \gamma(\tau)\lambda_{k,l} - f(\tau)d\tau\right)\right]_{k,l}\right)\right)\boldsymbol{w}_s$$

To obtain the solution to the original ODE (7), we take the inverse Fourier transform:

$$\boldsymbol{u}_t = \mathcal{F}^{-1}(\boldsymbol{w}_t) = \mathcal{F}^{-1}\operatorname{diag}\left(\operatorname{vec}\left(\left[\exp\left(\int_s^t \gamma(\tau)\lambda_{k,l} - f(\tau)d\tau\right)\right]_{k,l}\right)\right)\mathcal{F}\boldsymbol{u}_s$$

Hence, the transition operator for the original ODE (7) is:

$$\boldsymbol{A}(t,s) = \mathcal{F}^{-1}\text{diag}\left(\text{vec}\left(\left[\exp\left(\int_s^t \gamma(\tau)\lambda_{k,l} - f(\tau)d\tau\right)\right]_{k,l}\right)\right)\mathcal{F}$$

By (6.7) in Särkkä & Solin (2019), we know the mean of the transition density $p_{st}(\boldsymbol{u}_t \mid \boldsymbol{u}_s)$ is $\boldsymbol{A}(t,s)\boldsymbol{u}_s$.

Also by (6.7) in Särkkä & Solin (2019), we have:

$$\boldsymbol{P}(t,s) = \int_s^t g(\tau)^2 \boldsymbol{A}(t,\tau)\boldsymbol{A}^\top(t,\tau)d\tau \tag{8}$$

Since $\boldsymbol{A}$ is real, we know $\boldsymbol{A}^\top(t,\tau) = \boldsymbol{A}^*(t,\tau)$ where $\boldsymbol{\Psi}^*$ denotes its Hermitian adjoint. We can choose $\mathcal{F}$ and $\mathcal{F}^{-1}$ to be unitary operators, so that $\mathcal{F}^{-1} = \mathcal{F}^*$. Then we have:

$$\boldsymbol{A}^\top(t,\tau) = \boldsymbol{A}^*(t,\tau) = \mathcal{F}^{-1}\text{diag}\left(\text{vec}\left(\left[\exp\left(\int_\tau^t \gamma(r)\lambda_{k,l} - f(r)dr\right)\right]_{k,l}\right)\right)\mathcal{F}$$

and thus:

$$\boldsymbol{A}(t,\tau)\boldsymbol{A}^\top(t,\tau) = \mathcal{F}^{-1}\text{diag}\left(\text{vec}\left(\left[\exp\left(2\int_\tau^t \gamma(r)\lambda_{k,l} - f(r)dr\right)\right]_{k,l}\right)\right)\mathcal{F}$$

As integration is linear, we may bring $\mathcal{F}^{-1}$ and $\mathcal{F}$ outside the integral in Equation 8 and obtain:

$$\boldsymbol{P}(t,s) = \mathcal{F}^{-1}\text{diag}\left(\text{vec}\left(\left[\int_s^t g(\tau)^2 \exp\left(2\int_\tau^t \gamma(r)\lambda_{k,l} - f(r)dr\right)d\tau\right]_{k,l}\right)\right)\mathcal{F}$$

## C   EXPERIMENT DETAILS

We used a U-Net (Ronneberger et al., 2015) to represent the score $s_{\boldsymbol{\theta}}$. Our architecture used 128, 256, 512, and 1024 channels respectively at each resolution. Group Norm (Wu & He, 2018) was used after each convolution, followed by the Swish activation function (Ramachandran et al., 2017). The time $t$ is input to the score network using the Transformer's sinusoidal embedding (Vaswani et al., 2017), where the frequencies are randomly drawn from a Gaussian distribution with mean zero and standard deviation 30.

We train using a batch size of 256 and the Adam optimizer (Kingma & Ba, 2014) with a learning rate of $10^{-4}$. We diffuse using the heat-inspired SDE in Equation 4 using constant functions for $f$, $g$, and $\gamma$. We use $f(t) = -\log(0.01)$, $g(t) = 0.5$, and $\gamma(t) = 1$. These choices allow us to compute the frequency transition matrix $\boldsymbol{\Psi}(t,0)$ (see Equation 5) using the known antiderivative of a constant function. We compute the frequency covariance $\boldsymbol{\Sigma}(t,0)$ (see Equation 6) using the trapezoidal rule with a step size of $10^{-3}$. We train for 100 epochs.

