# OpenReview forum: "Diffusion Models in Space and Time via the Discretized Heat Equation"
_ICLR.cc/2022/Workshop/DGM4HSD — Submitted to ICLR 2022 DGM4HSD workshop_

### Official Review · Reviewer_MugB · 2022-03-21
**Interesting idea but presentation can be improved**

**Rating:** 4
**Confidence:** 4

**Review:**

** Summary **

The submission proposes to modify the *encoder* part of a denoising diffusion model by extending the usual addition of scaled isotropic Gaussian noise with a Laplacian contribution. As a result, the diffusion happens over space and time simultaneously, in contrast to the usual approach using only time-dependent noise schedules. The method can be derived from the heat equation, a partial differential equation modeling the spatiotemporal evolution of heat in a medium.

There are some critical steps, which the author address in order to realize this approach:
1. the diffusion equation must be linear in order to derive the closed-form of the transition density as a multivariate normal density.
2. a naive computation of the mean and covariance of this density would be expensive. The authors approach this by leveraging the diagonalization of the Laplacian in frequency space, which leads to finding diagonal matrices instead of full matrices.
3. the method is instantiated for images, which are considered 2D lattices. Then the Laplacian can be approximated using the usual finite difference approach.

The authors use their model to estimate the scores of the corresponding reverse diffusion process and use it to generate samples of a model trained on the MNIST dataset.

** Discussion **

The idea of a spatiotemporal coupling in the encoder is interesting and to my knowledge novel. It fits the workshop well and could be of interest to many researchers in this domain. However, I am not sure about the clarity of the presentation and whether I see claims backed by evidence so far.

The derivation looks sound but the presentation could improve a lot from clarity - especially for readers who do not have a strong background in stochastic processes or physics.

While the motivation by the heat equation can serve some intuition for researchers with a background in the physical/engineering sciences, it is not necessary to understand the approach. Together with domain-specific jargon, this can lead to confusion for machine learning researchers with other backgrounds. Examples are "encourages the dispersal of 'heat' in an image", "time-dependent thermal diffusivity". As the method is currently tied to images anyways due to the discrete Laplacian approximation a more natural motivation in this context could be derived from classic image processing techniques.

The paper could clarify more clearly at which points results are the correct solutions or where necessary approximations have to be made. For example, what is the impact of the discretization of the Laplacian to the final results? In algorithm 1 and the implementation, given for the covariance: how does the numerical integration affect results? How scalable is this?

Additionally, the paper contains many assertions for which no sources are cited or no evidence is provided. E.g., on the first page, the method is motivated by the statement `The more smoothly the noising process interpolates between the data distribution and the noise distribution, the easier it is to accurately estimate the score, yielding benefits in training time, synthesis
quality, and sampling speed"`, which is not further backed by any support.

Furthermore, I think the evaluation is insufficient - even for a workshop - at this point and should be elaborated on in a revision. If the authors motivate their approach by an *improved* encoder taking into account spatio-temporal relationships there should be at least some experimental evidence given in the results. The mere fact that the model can sample digits (without providing any quantitative statements, e.g., like a likelihood estimate), makes it unclear for a reader whether the method can help or not. I am not asking for SOTA results - but at least some indication that backs the claims leading to the initial motivation.

Finally, (this is minor relative to the other concerns) a holistic discussion of limitations of the method as well as a more comprehensive placement into the body of relevant literature would be appropriate. E.g., the very relevant prior work [Song 2021] is not cited.

** Recommendation **

I like the idea and I think it is worth discussing. However, at the current stage, I do not think that this manuscript is ready yet. If the authors address my points above (especially those about backing claims, at least some experimental validation) I would change my mind and recommend accepting it.

---

[Song 2021] Yang Song et al. Maximum Likelihood Training of Score-Based Diffusion Models, NeurIPS 2021.

---

### Official Review · Reviewer_P5UL · 2022-03-25
**This paper studies a new diffusion model by introducing a discrete Laplacian term to the drift of the diffusion. The theoretical formulas of the algorithm are presented with an empirical experiment on MNIST Dataset.**

**Rating:** 5
**Confidence:** 3

**Review:**

Detailed comments:

a.) This paper introduces a new Laplacian term to the diffusion model, the motivation is to solve the potential drawback of the traditional diffusion model that treat the space homogeneously. The idea is interesting, however, from formula (5) and (6), the stationary distribution may explore if the eigenvalues are not negative enough. More theoretical analysis is required.

b.) The empirical results are not so strong, we can clearly see the noise pixels around the generated images.

---

### Official Review · Reviewer_E6Xp · 2022-03-25
**Review of Diffusion Models in Space and Time via the Discretized Heat Equation**

**Rating:** 5
**Confidence:** 3

**Review:**

Goals: This paper aims at proposing a novel method for solving the inverse heat equation via machine learning.

Description: The authors manage to deliver their methods well. Via detailed derivation, the mathematical foundation is clear. However, lack of summary part as well as too brief description on experiment details makes it hard for audience to fully understand the methods proposed. Besides, the experiment is too simple to be viewed as a realistic scenario.

Evaluation: Again, lack of summary part makes the evaluation incomplete. The only evaluation for this method is discussed in the experiment section. However, with only 1 figure generated (Figure. 2) it is hard to reach a conclusion in terms of model evaluation. In the meantime, too simple experiment setup weakens the result. Based on the materials presented in this paper, it is hard to conclude whether the proposed method can be applied for a significant problem.

Clarity: The writing of this paper is clear, while to understand the content may require several background knowledge for stochastic analysis as well as partial differential equation. At mean time the paper doesn’t include a detailed discussion on its contribution. It only includes a simple example experiment which is not sufficient for evaluation.

Recommendation: The novelty of the proposed method is worth an acceptance. However, the paper itself should be improved by supplying more details as well as offering a concrete summary.

---

### Decision · Program_Chairs · 2022-03-26

**Decision:**

Reject

**Comment:**

All reviewers agree that the proposed idea is novel and worth discussing. However, the consensus is that the evaluation is very limited. It is not a-priori clear why the proposed diffusion process would be superior to the more conventional one without the Laplacian term, and therefore a comparison (qualitative or quantitative) is necessary, even for a workshop contribution. We encourage the authors to take into account such a comparison and resubmit to a future venue.